# Infused ice can multiply IceCube's sensitivity

Imre Bartos[1,2], Zsuzsa Marka[2] & Szabolcs Marka[2]

The IceCube Neutrino Observatory is the world's largest neutrino detector with a cubic-kilometer instrumented volume at the South Pole. It is preparing for a major upgrade that will significantly increase its sensitivity. A promising technological innovation investigated for this upgrade is wavelength shifting optics. Augmenting sensors with such optics could increase the photo-collection area of IceCube's digital optical modules, and shift the incoming photons' wavelength to where these modules are the most sensitive. Here we investigate the use of IceCube's drill holes as wavelength shifting optics. We calculate the sensitivity enhancement due to increasing the ice's refractive index in the holes, and infusing wavelength-shifting substrate into the ice. We find that, with adequate wavelength-shifter infusion, every ~0.05 increase in the ice's refractive index will increase IceCube's photon sensitivity by 100%, opening the possibility for the substantial, cost-effective expansion of IceCube's reach.

[1] Department of Physics, University of Florida, PO Box 118440 , Gainesville, FL 32611-8440, USA. [2] Columbia Astrophysics Laboratory, 550 West 120th Street, New York, NY 10027, USA. Correspondence and requests for materials should be addressed to I.B. (email: imrebartos@ufl.edu)

Since 2013, IceCube has been observing a diffuse flux of high-energy (TeV–PeV) neutrinos of cosmic origin[1,2]. The astrophysical sources producing these neutrinos are currently unknown. Their identification and detailed study will require a substantially expanded instrument. This motivated IceCube to plan a major upgrade, named IceCube-Gen2, over the next years to enhance its sensitivity to point sources by a factor of 5 and beyond[3]. This could be sufficient to resolve the sources of the observed cosmic neutrino flux[4,5], opening the door to a range of promising multimessenger observations. Additionally, IceCube will be extended with a low-energy (GeV) detector array, called PINGU[6], aiming to probe fundamental physics questions, such as the neutrino mass hierarchy, and could probe astrophysical GeV-neutrino sources[7].

Wavelength shifting (WLS) materials are a major consideration to achieve substantial and cost-effective increase in detector sensitivity[8–10]. Two key benefits of extending IceCube's digital optical modules (DOMs) with WLS are that light can be shifted from UV wavelengths of Cherenkov radiation to visible wavelength at which DOMs are the most sensitive; and WLS components can be added to collect and concentrate light, increasing sensitivity.

Here we explore the possibility to deploy WLS and concentrate light on the largest scales by effectively turning the drill holes in which strings of DOMs are submerged into WLS extensions. We consider two changes: increase of the refractive index of the drill hole ice; and deposition of WLS material into some or all of the drill hole ice (see Fig. 1 for illustration). In the following, we

investigate how the sensitivity of DOMs can be enhanced by these changes.

## Results

**Drill hole with high refractive index.** First, we examine the effect of an increased refractive index of the ice within IceCube's drill holes. We characterize the sensitivity of IceCube by the photon flux that reaches IceCube's DOMs for a fixed photon flux outside of the drill holes. In the following, we will refer to this measure as "DOM sensitivity". We average the results over the whole sky, equally weighting all directions. Separate results for the northern or southern sky are identical to this average over the whole.

Let DOMs with diameters $d_{DOM} = 0.25$ m be located along a vertical string within a drill hole with diameter $d_{hole} = 0.6$ m, vertically separated from each other by inter-DOM distance $h_{DOM} = 17$ m, following current IceCube parameters. Let the refractive index inside and outside the hole be $n_i$ and $n_o = n_{ice} = 1.32$[11], respectively. Here and in the following, indices i and o refer to parameters inside and outside the hole, respectively.

We are interested in the paths of light beams incident on the surface of the drill hole. We calculate the fraction of such light beams that reach within $d_{DOM}/2$ the hole's axis as a function of the hole's refractive index. This fraction is characteristic of the effective photo-collection area of the DOM-hole system.

In the following, we will adopt a Cartesian coordinate system with the drill hole oriented along the $z$ axis. Let a light beam reach the surface of the drill within the xz plane, at angle $\varphi_o$ from the $x$

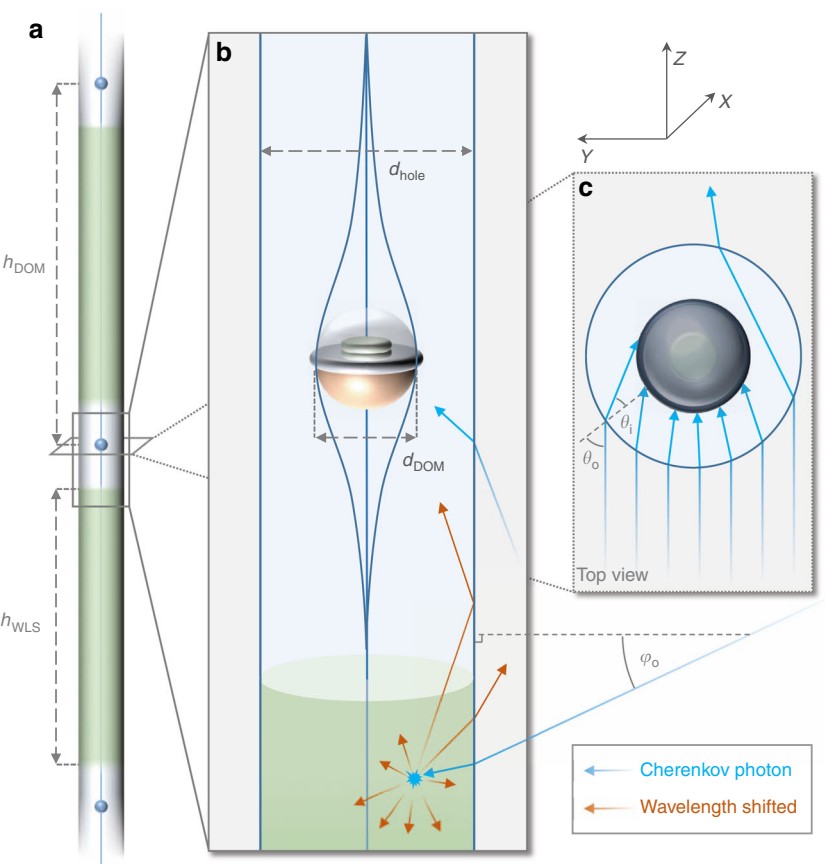

**Fig. 1** Illustration. **a** Illustration of a drill hole partially filled with infused ice with high refractive index and WLS. Also shown are the locations of optical modules connected by a wire. **b** Schematic drill hole in the vicinity of an optical module. Also shown is an example light propagation path. Light enters the hole at vertical angle $\varphi_o$, gets absorbed and re-emitted at a different wavelength. For large vertical re-emission angles, it gets reflected from the drill hole surface. Light entering the part of the hole with no WLS can directly reach the DOM. **c** Schematic refraction of light entering the drill hole with increased refractive index, shown in the vicinity of a DOM

axis. See Fig. 2 for illustration. The direction of the incoming light beam can be then described by its normal vector $\hat{\mathbf{N}}_o = (\cos\varphi_o, 0, \sin\varphi_o)$. Let $\hat{\mathbf{N}}_s = (-\cos\theta_o, -\sin\theta_o, 0)$ be the normal vector of the hole surface at the point of incidence, where $\theta_o$ is the angle between $-(\hat{\mathbf{N}}_o)_{xy}$ and $\hat{\mathbf{N}}_s$, where $()_{xy}$ denotes a vector's projection into the $xy$ plane. The normal vector of the refracted light beam inside the hole will be denoted with $\hat{\mathbf{N}}_i$. Finally, let the angle between $-\hat{\mathbf{N}}_o$ and $\hat{\mathbf{N}}_s$ be denoted with $\alpha_o$, and the angle between $\hat{\mathbf{N}}_i$ and $-\hat{\mathbf{N}}_s$ be denoted with $\alpha_i$.

With these definitions, we now calculate $\theta_i$, which is the angle between $(\hat{\mathbf{N}}_i)_{xy}$ and $-\hat{\mathbf{N}}_s$. This angle is necessary to determine whether the beam approached the central line of the drill hole within $d_{DOM}/2$. Separating the unknown components of $\hat{\mathbf{N}}_i$ as $\hat{\mathbf{N}}_i = (x_i, y_i, z_i)$, we can write

$$\hat{\mathbf{N}}_i \cdot \hat{\mathbf{N}}_s = \cos(\alpha_i) = -x_i\cos\theta_o - y_i\sin\theta_o, \tag{1}$$

$$\hat{\mathbf{N}}_i \cdot \hat{\mathbf{N}}_o = \cos(\alpha_o - \alpha_i) = x_i\cos\varphi_o + z_i\sin\varphi_o, \tag{2}$$

where, for the second equation, we use the fact that refraction ensures that $\hat{\mathbf{N}}_i$, $\hat{\mathbf{N}}_o$, and $\hat{\mathbf{N}}_s$ are all within the same plane. We solve this equation system analytically, with the additional constraint $x_i^2 + y_i^2 + z_i^2 = 1$, to obtain $x_i$, $y_i$, and $z_i$. We find $\theta_i$ by writing

$$\hat{\mathbf{N}}_i \cdot \hat{\mathbf{N}}_s = \sqrt{x_i^2 + y_i^2}\cos(\pi + \theta_i) = -\sqrt{x_i^2 + y_i^2}\cos(\theta_i). \tag{3}$$

Combining Eqs. (2) and (3) gives

$$\theta_i = \arccos\left(\frac{x_i\cos\theta_o + y_i\sin\theta_o}{\sqrt{x_i^2 + y_i^2}}\right). \tag{4}$$

We numerically calculate the width $W(\varphi_o, n_i)$ of the part of the incoming light beam outside the drill hole that will reach the central cylinder in the drill hole with diameter $d_{DOM}$. This width can be determined by requiring its edges to satisfy $\sin\theta_{i,W} = d_{DOM}/d_{hole}$ (see Fig. 2). To good approximation, $W(\varphi_o, n_i) \propto n_i$ until it reaches $W = d_{hole}$. For the case of $\varphi_o = 0$, we analytically find the relation $W(\varphi_o = 0, n_i) = d_{DOM} n_i n_o^{-1}$.

We then obtain the total flux $F_n(n_i)$ of incoming light reaching the inner cylinder for inner-hole refractive index $n_i$ by integrating this width over all vertical angles. The resulting total flux $F_n(n_i)$, normalized by the baseline case of no increased refractive index ($n_i = n_o$), is

$$\frac{F_n(n_i)}{F_n(n_o)} = \frac{1}{2}\int_{-\pi/2}^{\pi/2} \frac{W(\varphi_o, n_i)}{d_{DOM}}\cos(\varphi_o)\,d\varphi_o. \tag{5}$$

The obtained fractional increase in the incoming flux, $F_n(n_i)/F_n(n_o)$, is shown in Fig. 3 as a function of $n_i$. We see that 0.1 increase in $n_i$ corresponds to ~10% flux increase onto IceCube's DOMs.

**Wavelength shifters.** We investigate the enhancement of the detected photon flux due to the infusion of ice in the drill holes with WLS. WLS can enhance sensitivity by absorbing UV photons (200–350 nm) and re-emitting them in the visible range where DOMs are most sensitive (~400 nm; see ref. [8], in particular their Fig. 3). UV photons are more abundant in Cherenkov radiation than visible since the Cherenkov spectrum is $\propto \lambda^{-2}$, where $\lambda$ is the photon wavelength.

For the following calculations, we assume that WLS has the following properties: incoming UV light that enters WLS will be fully absorbed and re-emitted at visible wavelength, i.e., the efficiency of conversion for photons entering the WLS region is 100% (cf. [9]); additionally, IceCube's DOMs will be assumed to have $\epsilon_{eff} = 40\%$ higher light detection efficiency at the re-emission wavelengths compared to the original UV[8]; furthermore, absorbed photons will be re-emitted isotropically, independently of the direction of the incoming beam.

Wavelength-shifted light will be bound to within the drill hole by total internal reflection (tir) if it is re-emitted at an angle that is smaller than the critical angle. Therefore, a fraction $f_{tir} = 1 - n_o/n_i$ of the re-emitted light will be totally internally reflected, staying within the drill hole.

We next consider the effect of absorption and scattering in the ice. Scattering length deep in the ice at the South Pole at 400 nm is comparable or greater than $h_{DOM}$,[12] while the absorption length is typically significantly greater than $h_{DOM}$, varying from

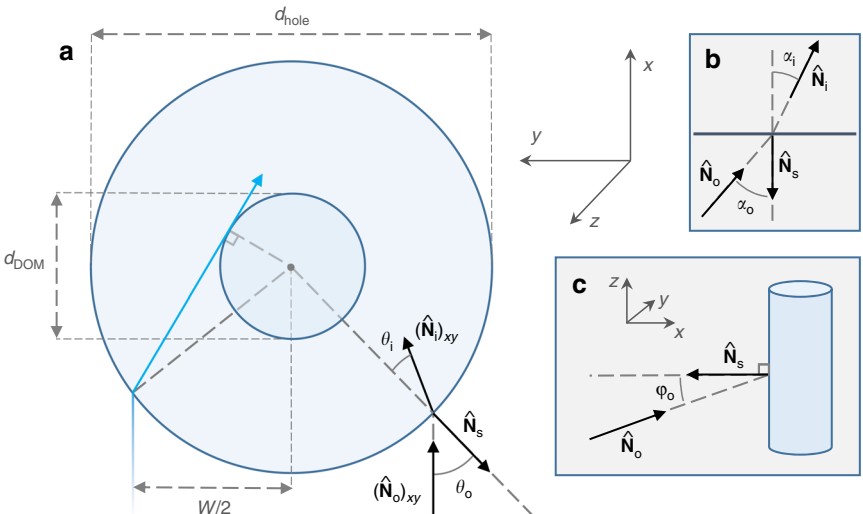

**Fig. 2** Notation for refraction calculation. Illustration for the notations of angles, normal vectors, and sizes used in the calculation. **a** Schematic top view of a drill hole, showing the DOM in the center. **b** View in the plane of beam propagation. **c** Side view from a horizontal direction. For all three views, we show a subset of the normal vectors of an incoming light beam $(\hat{\mathbf{N}}_o)$, the refracted light beam $(\hat{\mathbf{N}}_i)$, and the normal vector of the hole surface $(\hat{\mathbf{N}}_s)$. The index $()_{xy}$ indicates that the projections of the vectors are shown to the $xy$ plane. Also shown are the angles between the shown vectors

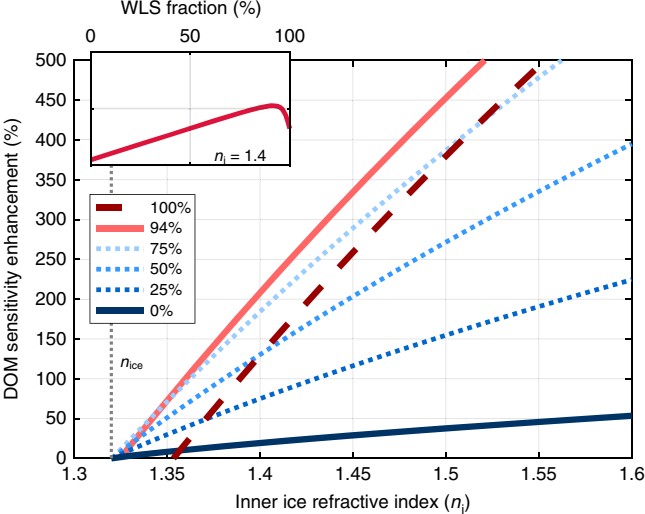

**Fig. 3** DOM sensitivity enhancement as a function of the inner ice refractive index. Fractional enhancement of DOM sensitivity due to the increase of the ice refractive index within drill holes and the infusion of wavelength-shifting material, as a function of the inner ice's refractive index, for different fractions of the drill hole infused with wavelength-shifting material (see legend). The case of no wavelength shifter corresponds to 0% filling fraction

50 m to 150 m for light from UV to blue, depending on depth[13,14]. We conservatively approximate the effect of absorption and scattering using the following simple model. We assume that photons entering the drill hole between DOMs $i$ and $j$ can only be detected by either DOM $i$ or DOM $j$. That is, they cannot travel past a DOM, which would mean a path $\gtrsim h_{DOM}$. For simplicity, we assume that re-emitted light within the drill hole will have a uniform vertical flux distribution, and only those photons hitting a DOM, which is a fraction of $d_{DOM}^2/d_{hole}^2$ of the total flux, will contribute to detection.

For light that would be reaching the DOMs without the presence of WLS, it is more efficient to not place WLS in its way, as isotropic re-emission can redirect a significant fraction of it. Therefore, while it is important to increase the refractive index of ice everywhere in the drill hole, it is optimal to only partially infuse the ice with WLS (see Fig. 1). We consider infusing ice with WLS within a vertical height $h_{WLS}$ of the total height $h_{DOM}$ available between DOMs. An infused volume with height $h_{WLS}$ will have an effective area $A_{WLS} = h_{WLS}d_{hole}\cos\varphi_o + \pi/4 d_{hole}^2 \sin\varphi_o$, where $\varphi_o$ is the vertical angle corresponding to the direction of the Cherenkov source. For a photon flux $F$ at the detector, the rate $R$ of photons entering the infused ice between two DOMs will be $R = FA_{WLS}$.

Integrating over all vertical contributions, we find that the total increase in flux onto a DOM, normalized by the baseline flux $F_n(n_o)$, is

$$\frac{F_{WLS}(n_i)}{F_n(n_o)} = \frac{1+\epsilon_{eff}}{2}\int_{-\pi/2}^{\pi/2} \frac{4A_{WLS}}{\pi d_{DOM}^2} f_{tir} \frac{d_{DOM}^2}{d_{hole}^2}\cos\varphi_o \mathrm{d}\varphi_o. \quad (6)$$

We note here that light absorption by the WLS will have negligible effect on the flux reaching more distant DOMs. Taking a string spacing of 240 m and a string length of 1 km, for any given DOM a neighboring drill hole of 0.6 m diameter covers less than $5 \times 10^{-4}$ of its view, making the coverage even from all holes negligible.

**Contribution from hole with no wavelength shifter**. The part of the drill hole that is not infused with WLS will contribute to the DOM flux only due to the increased refractive index. To obtain this contribution, we employ Eq. (5), with the modification that we integrate only over vertical angles that do not cross WLS material. The maximum vertical angle for which a light beam heading toward a DOM does not cross WLS can be approximated as $\varphi_{o,max} \approx \arctan[(h_{DOM} - h_{WLS})/d_{hole}]$. The fractional increase of the photon flux at the DOM due to the part of the hole with no WLS can be calculated using Eq. (5), but with modifying the interval of the integral on the right side to $[-\varphi_{o,max}, \varphi_{o,max}]$.

**DOM sensitivity enhancement**. We calculated the fractional enhancement of the DOM sensitivity due to infused ice by combining the contributions from WLS material and from the part of the drill hole that is not infused with WLS. We considered different $h_{WLS}/h_{DOM}$ WLS filling factors. See Fig. 3 for DOM sensitivity enhancement for different filling factors and refractive indices. We found that optimally 94% of the drill hole should be filled with WLS, leaving $0.06h_{DOM} \cong 1$ m WLS-free around DOMs. It is also clear though that the results are not overly sensitive to this precise value of the filling factor, making it easier to achieve adequate partial filling, without needing to precisely target 94%.

The above results shown in Fig. 3 present an aggregated result averaged over the sky. The enhancement of DOM sensitivity will nevertheless depend on the source orientation. We explore the direction-dependence of sensitivity enhancement in the Methods section (Direction-dependent DOM sensitivity enhancement).

**Photon temporal delay**. Beyond enhanced photon sensitivity, WLS will also introduce a delay in the time of arrival of photons to DOMs. This will affect signal reconstruction in which photon timing is one of the most important factors (e.g., ref. [15]).

The time delay due to WLS re-emission will mainly come from the fact that Cherenkov photons interact with the WLS material away from the DOM. The added delay due to absorption and re-emission by the WLS is small (~2 ns[9]). The re-emitted photons then need to travel to the DOM. The characteristic delay of these re-emitted photons compared to photons that directly reach the DOMs can be written as

$$\mathrm{d}t_{char} \sim \frac{h_{DOM}n_i^2}{cn_o} \approx 84\,\mathrm{ns}\left(\frac{n_i}{1.4}\right)^2, \quad (7)$$

where we took the critical angle $\theta_c$ of the hole surface to be the characteristic direction of photons, elongating the track by a factor of $1/\sin\theta_c = n_i/n_o$. As we show in the Methods section, a simple uniform time shift of $\mathrm{d}t_{char}$ accurately characterizes the effect of infused ice on timing for Cherenkov-source distances $\gg h_{DOM}$ from the DOM.

**Next steps**. Beyond the general description above, increasing IceCube's sensitivity will depend on the specifics of the implementation. Below we discuss some of these challenges.

Probably the most important question is the composition and concentration of the additives that will modify the refractive index and enable WLS. This choice will determine the level of improvement in sensitivity, but cost and environmental considerations will also be critical. We discuss some feasible choices and challenges with additives further in detail in the Methods section.

Increasing the refractive index of the hole ice will also modify the difference in refractive index between the hole ice and the DOMs' pressure vessel. This will decrease the focusing properties of the DOMs' glass, potentially decreasing sensitivity. This effect

needs to be understood and taken into account in the design of new DOMs.

For WLS, we used the simplifying assumption that essentially all photons are wavelength shifted and isotropically re-emitted. Adopting WLS material that shifts photons only at wavelength below the sensitive range of the DOMs, we could essentially treat the direct detectable light and the wavelength-shifted light independently. This would enable a continuous filling of the hole with WLS material as it would not block the path of detectable photons. This would simplify the infusion process and may further improve sensitivity.

The feasibility of uniformly infusing the ice in the drill hole with a WLS material such as those described in ref. [8] will also need to be investigated. The required concentration of WLS is small, likely in the mg L$^{-1}$ range or below, due to the increased probability of light absorption in the large size WLS infused volume.

We note here that the new Rapid Access Ice Drill (RAID)[16] project's boreholes (up to 3300 m deep) are to be kept open using antifreeze drilling fluid as long as possible, and thus possibly available for future down-hole measurements. RAID may provide an opportunity for the above-mentioned experimental investigations.

We further need to study the effect of uneven drill hole surface; diffusion of phase-shifted light within the drill hole; possible corrosion due to the soluble; and the feasibility of carrying the needed materials to the South Pole.

It will be worth investigating the possibility of modifying the properties of ice beyond the drill hole by expanding the hole, e.g., around DOMs. Additionally, one can alter ice at locations other than drill holes, e.g., by placing additional holes within the instrumented volume.

The IceCube-Gen2 design will likely feature updated DOMs with significantly improved photon detection efficiency, increased photo-collecting area, and possibly other improvements[3]. It is worth noting here that these changes can enhance the improvement in photon detection efficiency due to infused ice. For factors of $\kappa_{DOM}$ and $\kappa_{ice}$ improvements in photon detection efficiencies due to a DOM upgrade and infused ice, respectively, the combined improvement will be $\kappa_{DOM}\kappa_{ice}$, motivating the combined use of these two upgrades. For wavelength-shifting optical modules (WOMs[9]), the situation is somewhat different as here photons shifted in the infused ice will not be shifted again in the WOM, and they will also propagate differently, so the added benefit of infused ice to WOM modules will be more limited. It will be useful to investigate the interplay between infused ice and different DOM designs, as well as possible DOM design optimizations for combined use with infused ice.

The present work focuses exclusively on IceCube's upgrades, but other possibilities of using WLS additives are interesting to explore, such as their use in containers in water Cherenkov detectors, or as cost-effective scintillators at the South Pole.

IceCube's utility depends on many factors, including energy-dependent effective area, background rejection, as well as direction and energy reconstruction ability. The dependence of these factors on IceCube's DOM sensitivity will need to be explored using detailed simulations that incorporate the detector geometry, the effects of infused ice, and the reconstruction algorithms. In the Methods section, we discuss two examples of other sensitivity measures, and estimate their expected increase with improved DOM sensitivity. A particularly important challenge is understanding the role of temporal delay for re-scattered compared to direct light. Timing information is important in reconstructing the point of origin of Cherenkov emission by comparing time stamps in multiple DOMs. Scattered photons therefore carry less information than direct photons. An added ambiguity is that the time delay of re-scattered photons will be a measure of the Cherenkov-source distance from the infused drill hole, not the DOM itself. Furthermore, in general it will be difficult to differentiate direct and re-scattered photons, decreasing the information content of the prior. This effect will be mitigated by the fact that the earliest photons will be direct, and therefore will carry information on the distance of the Cherenkov source. On the positive side, the importance of the additional time delay becomes less important for more distant Cherenkov sources, which benefit the most from the increased photon count, while for nearby sources, there can be a sufficient direct photon flux to recover the source properties. The interplay of these factors, and the overall sensitivity of infused ice to different astrophysical sources, will require detailed simulations.

## Discussion

We explored the possible improvement in IceCube-Gen2's sensitivity due to infusing drill holes with high-refractive-index material and partially with appropriate WLS material. We find that the optimal configuration is 94% infusion with WLS material, for which we find that every 0.05 increase in the ice's refractive index adds 100% to IceCube-Gen2's DOM sensitivity. Most of this increase is due to the isotropic re-emission property of WLS.

While our calculations involved approximations and simplifying assumptions, it is clear that there is a potentially enormous benefit in modifying the properties of the ice in drill holes during the placement of DOM strings. We reviewed some of the important next steps in determining the proposed method's feasibility and developing a deployment plan. It will also be worth investigating whether large-scale placement of wavelength-shifting volumes could be beneficial in the case of water-based Cherenkov detectors.

## Methods

**Detailed photon temporal delay calculation**. In the Results section, we characterized the temporal delay of photons entering the infused ice compared to photons directly reaching the DOM by $dt_{char}$ (see Eq. (7)). Here we give a more detailed comparison and show that $dt_{char}$ accurately describes the overall delay for neutrino interaction distances from the DOM $\gg h_{DOM}$.

For simplicity, we assume a neutrino interaction that induces Cherenkov photons within a volume small compared to its distance from a given DOM, such that we can treat the Cherenkov source as a point source. We consider a coordinate system in which the DOM is in the origin, and the $z$ axis points upward. Let the location of the neutrino interaction be $\mathbf{r}$, at a distance $r = |\mathbf{r}|$ from the DOM.

First, we calculate the temporal distribution of Cherenkov photons directly reaching the DOM. The dominant effect on photon arrival times is scattering in the ice[15]. We describe the distribution of photon arrival times due to scattering analytically, using the Pandel function[17]. The Pandel function describes the distribution of residual time $t_{res}$ of photon arrivals compared to their un-scattered time $rn_{ice}/c$. It is an analytical treatment of scattering with parameters empirically determined using Monte Carlo simulations of light scattering in ice. Here we adopt the description and parameters of ref. [15] (see their Section 3.2.1). Results for the probability distribution of $t_{res}$ for $r = 100$ m and $r = 20$ m are shown in Fig. 4. We see that the characteristic residual time due to scattering over 100 m is ~400 ns, with considerable spread. For the case of $r = 20$ m, the distance is much smaller than the characteristic scattering length, making the characteristic residual delay small for photons directly propagating to the DOM.

Next, we use the following simulation to estimate the probability distribution of time delays expected for the infused ice. We first consider a thin layer of the drill hole infused ice located at $z_{WLS}$ vertical coordinate. The distance of this layer from the origin of Cherenkov photons is $r_{WLS} = |\mathbf{r} - (0, 0, z_{WLS})|$, where we expressed the location of the drill hole shell with Cartesian coordinates. We generate random residual times for photons reaching the drill hole shell from the Cherenkov source using the Pandel function for distance $r_{WLS}$. Each photon is then randomly oriented with uniform directional distribution. We denote the angle of this new direction with $\varphi_{i,shifted}$, with $\varphi_{i,shifted} = 0$ corresponds to the horizontal direction. Photons that will not internally reflect on the drill hole wall $\left(|\varphi_{i,shifted}| < \theta_c\right)$ are discarded. For the remaining photons, an additional time delay of $z_{WLS}n_i c^{-1}(\sin\varphi_{i,shifted})^{-1}$ is added, which is their travel time to the DOM following re-emission. The

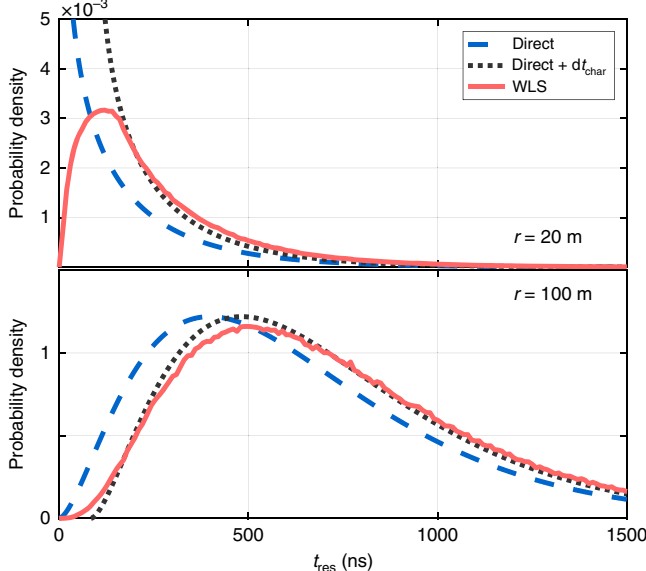

**Fig. 4** Cherenkov photon time delay distribution. The distribution of residual time delays ($t - rn_{ice}/c$) are shown for photons propagating to the DOM from vertical angle $\varphi_o = 45°$, from distances $r = 20$ m (top) and $r = 100$ m (bottom). Shown are the distributions of residual delay for photons propagating directly to the DOM from the Cherenkov source (dashed line), which is described by the Pandel function. Also shown are the distributions of simulated delays for photons that first enter the infused ice, get wavelength shifted, and then reach the DOM by propagating within the drill hole (solid line). For comparison, we show the distributions for direct propagation modified by the characteristic delay $dt_{char}$ due to infused ice (dotted line; see Eq. (7))

total residual delay for a given photon $i$ is therefore

$$t_{res,WLS,i} = r_{WLS} n_o c^{-1} + t_{Pandel,i}(r_{WLS}) + z_{WLS} n_i c^{-1}\left(\sin\varphi_{i,shifted}\right)^{-1} - rn_o c^{-1}, \quad (8)$$

where $t_{Pandel,i}(r_{WLS})$ is randomly drawn from the Pandel function at distance $r_{WLS}$.

In order to account for the vertically changing flux of Cherenkov photons as a function of $z_{WLS}$, we analytically approximate the photon flux as a function of $r_{WLS}$. We adopt the approximating formula of ref. [18] (see their Section 3.1) that takes into account absorption and scattering, using characteristic absorption and scattering length $\lambda_a = 98$ m and $\lambda_s = 24$ m, respectively. We use this approximate flux in weighing the contribution of different drill hole shells compared to each other in the overall delay distribution.

We finally integrate over the thin drill hole shells in the interval $z_{WLS} \in [-h_{DOM}, 0 \text{ m}]$ in order to obtain the overall distribution of time delays for wavelength-shifted photons. Representative results are shown in Fig. 4 for a Cherenkov point source at distance $r = 100$ m with $\varphi_o = 45°$.

We see in Fig. 4 (bottom) that the difference between the residual time distribution of direct photons to the DOM and those that were wavelength shifted can be well characterized by a uniform time shift of $dt_{char}$ (see Eq. (7)) for every photon, at least for the $r = 100$ m considered. Carrying out the same calculation for different distances, we find that this single-shift characterization is adequate for $r \gg h_{DOM}$. For the case of $r = 20$ m $\approx h_{DOM}$, shown in Fig. 4 (top), the characterization with a uniform time shift only works at late times.

**Additives—options and next steps**. The choice of additives will be pivotal in the prospects of detector improvement with infused ice. As DOM sensitivity is strongly dependent on the ice's refractive index, it will benefit from a more effective or higher-concentration additive. At the same time, the feasibility of transporting large quantities of additives, as well as potential environmental effects, must be considered. Below we briefly outline some of the possibilities and foreseeable challenges that need to be further investigated.

A critical point is identifying suitable additives for increasing the holes' refractive index. As, to our knowledge, there are no detailed studies under high-pressure, low-temperature conditions, we outline some findings for more standard conditions (e.g., refs. [19,20]). For example, for aqueous solutions at room temperature, $n = 1.38$ refractive index at 589 nm can be achieved with 26% NaCl or 30% sucrose solution[20].

Some high-concentration (30%) solutions were also shown to markedly increase the refractive index even at low temperatures[21].

A potential drawback for some of these additive materials can be their radioactivity. For example, sea salt is radioactive primarily due to $^{40}K$, which is responsible for $O(10^4)$ kHz background rate in water-based Cherenkov detectors[22]. Organic materials are also radioactive at a similar levels due to $^{14}C$. At the same time, some other available salts, such as $MgCl_2$, have essentially no radioactivity. Salt purification is another possibility. Further studies on the refractive index of solute infused ice are necessary, including temperature and pressure dependence.

Some additives could be additionally useful to enhance neutron detection by IceCube via higher cross-section and greater light emission. This may merit the addition of distinct solubles, but further studies are necessary.

Aqueous solutions typically undergo fractional freezing during which the pure solvent freezes, concentrating dissolved impurities at the boundaries of the frozen regions. This complicates the modification of the ice's refractive index, and possibly the distribution of wavelength shifters. Nevertheless, homogeneous ice nucleation in aqueous solutions in some cases is possible[23,24]. Upon cooling, both crystallization and vitrification (glassy solid) behavior were observed depending on solute concentration. Another important factor is the cooling rate. For example, at as low as 0.1 K min$^{-1}$ cooling rate, homogeneous glassy solid was observed forming from a wide range of concentrations of highly concentrated LiCl solutions[25]. The homogeneity of the freezing process for aqueous solutions of concentration range of interest, for volumes corresponding to a realistic vertical segment of a drill hole must be investigated as available information is based on small volume investigations. Another possibility is using an antifreeze agent in the drill hole. For example, for 50% ethylene glycol in water $n = 1.38$ at 589 nm[20], while for amyl acetate $n = 1.4$ at 589 nm and 20 °C[20] (freezes at −71 °C). Liquid holes will likely require active circulation.

The transportation of additives to the South Pole is a challenge that can affect the optimal choice of ingredients or the infused volume. Taking a fiducial 10% solution, one drill hole would require $\sim 0.1\pi d_{hole}^2 \times (1\,\text{km}) \approx 30$ tons of additive to be transported. For comparison, ~2800 tons of cargo and fuel are delivered annually to the South Pole on the ground (2010/11 data[26]). For IceCube-Gen2's high-energy extension with 120 strings, the total required weight is equivalent to 130% of the annual payload delivered to the South Pole. For IceCube's low-energy upgrade PINGU with 20 shorter (300 m) strings[6], the required weight is equivalent to 6% of the annual payload delivered, or one mission with an LC-130 aircraft per string.

Multiple directions will be explored to further mitigate the required amount of additives: additives that boost the refractive index at lower concentrations; improvement of transportation to the South Pole; and partial infusion. An example for the latter case is the application of additives only at the boundary of the detectors, where it can enhance our vetoing capability by helping differentiate tracks that originate inside vs. outside of the instrumented volume, or by helping to detect showers closer to the surface. These options need to be explored to optimize the cost benefit ratio.

**Sensitivity estimates**. IceCube searches for a large variety of sources, and its sensitivity strongly depends on the source type considered (e.g., ref. [27]). It is therefore difficult to characterize any improvement with a single measure. In this paper, we focused on DOM sensitivity—the number of photons detected per unit flux. This measure converts into other sensitivity measures in a complex manner, especially considering that increased DOM sensitivity may allow other changes, such as increased string spacing, as the optimal configuration. Here, as two examples, we describe the comparison of two PINGU detector designs and the conversion of DOM sensitivity to supernova-neutrino sensitivity.

First, we consider the case of IceCube's planned low-energy extension, PINGU[6], which focuses primarily on neutrino oscillation physics. During the design of PINGU, multiple detector geometries were considered. The evaluation metric between geometries was chosen to be the sensitivity to the neutrino mass ordering. Part of the motivation for choosing this sensitivity measure was that geometry-dependent quantities, such as the energy threshold or the energy and angular reconstruction resolution, which affect the determination of mass ordering are common features that affect other searches as well[6]. Detailed simulations of the detector and optimization of the geometry found that a configuration with 40 strings, 96 DOMs per string, and 22-meter string spacing showed comparable sensitivity to an updated baseline design with 26 strings, 192 DOMs per string, and 24-meter string spacing. We see in this example that an effective doubling of DOM sensitivity, by virtue of doubling the DOM density on each string, delivers comparable sensitivity to mass ordering requiring only 65% of the strings. This reduction can be critical for PINGU as it can allow drilling to be completed over fewer seasons, significantly reducing the construction cost.

Some caveats with this comparison are that further increase in effective DOMs sensitivity may not necessarily translate into further gains of "mass ordering" sensitivity, and that the above simulations do not factor in the added time delay from infused ice. In particular, a major factor in determining the neutrino mass hierarchy is the ability to distinguish events induced by muon neutrinos from those induced by electron or tau neutrinos (i.e., tracks vs. showers). Added time delay from WLS can adversely affect this ability. Nevertheless, we see that in this example the total DOM sensitivity per string had significant effect on other sensitivity measures.

As a second example, we consider the core-collapse of massive stars, which produces a strong burst of $\mathcal{O}(10\,\text{MeV})$ neutrinos with total radiated energy of

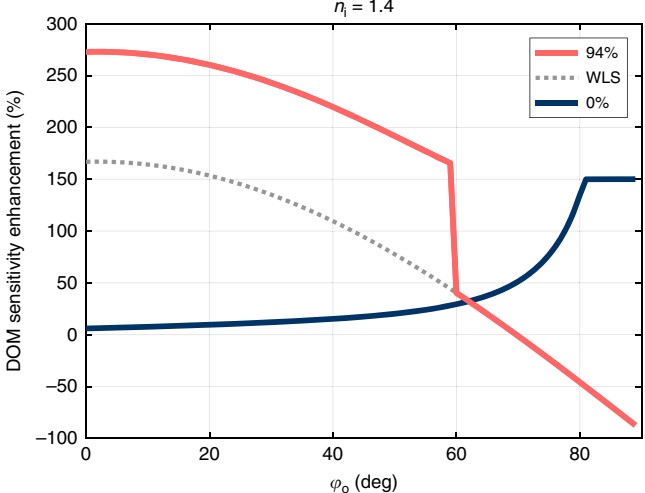

**Fig. 5** Direction-dependent DOM sensitivity enhancement. DOM sensitivity enhancement is shown as a function of the vertical angle of the incoming light beam ($\varphi_o$; see Fig. 2), for $n_i = 1.4$. The enhancement is shown for optimal WLS filling of 94% (red line) as well as for no WLS (0%; blue line). Also shown is the WLS contribution to the enhancement for the optimal case (dotted line); the non-WLS contribution is equivalent to the enhancement for no WLS below 59° above which the WLS blocks direct light to the DOM

~$10^{53}$ erg. IceCube can detect such a burst out to tens of kilo-parsecs[28]. While individual MeV neutrinos are not detectable, Cherenkov light induced by neutrino interactions will increase the total photon count rate in the DOMs. For sufficiently large increase compared to photomultiplier noise rate, the MeV neutrino flux is detectable.

The detection of a supernova neutrino burst relies on the identification of a statistically significant increase in the total photomultiplier rate of the detector over the burst time scale, which is $\gg$1 ms[28]. A consequence of this time scale is that the photon propagation time within the infused ice can be neglected here. For a detector with infused ice, the signal photon count rate will grow proportionally with the increase in DOM sensitivity. The noise is dominated by radioactive decay within the glass of the DOMs themselves, therefore the noise level will not be significantly changed by the infused ice. Therefore, for a fixed significance, infused ice increases the distance $d_{SN}$ from which one can detect a supernova through neutrinos by

$$\frac{d_{SN}(n_i)}{d_{SN}(n_o)} \equiv \left(\frac{F_{WLS}(n_i)}{F_n(n_o)}\right)^{1/2}, \qquad (9)$$

where the right side of the equation is the ratio from Eq. (6). We note that an enhanced detection range will not only potentially increase the detection rate, but will also provide greater resolution to observed supernovae that can help better understand the evolution of the core collapse[29].

**Direction-dependent DOM sensitivity enhancement**. Here we examine the dependence of DOM sensitivity enhancement on $\varphi_o$, i.e., the orientation of the Cherenkov source compared to the DOM. We use Eqs. (5) and (6) without the integral and weight (cos $\varphi_o$) on the right side of the equations.

Representative results for $n_i = 1.4$ are shown in Fig. 5. We see that, for optimal WLS filling (94%; see Fig. 3), we achieve the most enhancement for emission angles closer to horizontal. This is expected as the effective area of the WLS is greatest in these directions. It is interesting to see that the WLS blocks direct light from reaching the DOM at angles >59°, so for these directions we actually lose sensitivity. Figure 5 also shows results for the case of only increased refractive index but no WLS (0%). Here we see that we gain the most for more vertical directions, with a saturation at ~150%. This angle corresponds to the focusing of all photons entering the drill hole onto the central region.

It is important to note here that this directional-dependent sensitivity does not directly correspond to dependence on the direction of the neutrino. For neutrino interactions that produce cascade events, the location of the interaction can be anywhere in the detector independently of the neutrino direction, and primarily this location with respect to each DOM will determine the direction of the Cherenkov photons. For neutrino interactions that produce track events, the tracks induce Cherenkov light at ≈41° from the direction of the neutrino. Neglecting the change of photon directions due to scattering, this is the relevant characteristic direction to consider for vertical events, both up-going and down-going. By

comparing Figs. 3 and 5, we see that at 41°, DOM sensitivity enhancement in Fig. 5 is similar to the direction-averaged DOM sensitivity enhancement in Fig. 3. For track events that are not vertical, emission will occur at a range ±41° around the neutrino direction, making a more complex interplay between the neutrino direction and DOM sensitivity enhancement. We expect that more horizontal directions will be more important for this case as they will likely reach a drill hole over a much shorter path than the more vertical photons, making their flux higher. This preference to more horizontal photons is favorable given the higher DOM sensitivity enhancement in these directions.

**Data availability**. The data that support the findings of this study are available from the corresponding author upon request.

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

## Acknowledgements
The authors are grateful to Barry Barish, Ryan Bay, Segev BenZvi, Tyce DeYoung, Michael DuVernois, Chad Finley, Darren Grant, Francis Halzen, Dustin Hebecker, Alexander Kappes, Spencer Klein, Claudio Kopper, Martin Rongen, Joerg Schaefer, and Dawn Williams for their valuable comments and suggestions. The authors are thankful for the generous support of the University of Florida and Columbia University in the City of New York.

## Author contributions
I.B., Z.M., and S.M. each contributed to the origination of the idea for the project and worked out the general details collaboratively. I.B. carried out the calculations.

## Additional information

**Competing interests:** The authors declare no competing interests.

