## [Peer Review File(PDF 486 kb) · Nature Communications]

Reviewers' comments:

Reviewer #1 (Remarks to the Author):

The paper addresses a new development for a modification of the planned future upgrade of IceCube, the largest neutrino detector existing in ice. A significant enhancement of the sensitivity is claimed to be potentially reachable by seemingly straight forward modifications in the detector, which are elaborated on.

The topic is of high relevance for the future science potential of the planned expansion of the IceCube detector, which is currently the largest existing neutrino detector. The planned IceCube expansion will explore cosmic high energy neutrinos with unprecedented statistics and any significant boost of its sensitivity will directly translate to the science reach, which is of high value to the astroparticle physics community.

The paper addresses an interesting seemingly promising novel avenue for gaining sensitivity and deserves publication and further studies. Overall the paper contains a very interesting idea and considers the relevant important aspects, still there are a few points which I would like to see better clarified for the publication to support the notion of a large gain.

In order to enhance the sensitivity the authors consider the possibility to modify the ice in the deep holes, in which the detector strings are mounted. Hereby they focus on two effects: 1) An enhancement of the light yield by using wavelength shifting additives to transfer part of the before undetectable light to the wavelengths where the detectors are sensitive and also to then rescatter the light from the hole volume to the detectors. 2) An enhancement of the light yield by the change of the refractive index of the medium in the ice hole to focus the light hitting the hole onto the detectors.

While the change of the refractive index alone leads only to moderate gains the claim is that the wavelength shifting and rescattering together can already with small changes of the refractive index result in large gains in the light yield.

The technical challenge is also mentioned and some suggestions put forth for the actual realization of such a modified detector.

The paper contains a very interesting idea and considers the important aspects, still there are a few points which I would like to see better clarified for the publication to substantiate the claims.

The main gain in sensitivity is demonstrated to stem from the addition of wavelength-shifted light: The rescattering of the light passing the hole is in this way becoming detectable, but otherwise is assumed to be missed in the detection.

Here the paper does not provide guidance to the amount of information loss by rescattered light, which otherwise could actually have been detected elsewhere (I assume though very little with the sparse IceCube Gen2 design). It also does not elaborate how much the scattering compromises the information from the light which then is detected. It is only pointed out that effects on the reconstruction are not yet considered and deserve further detailed studies.

Even that indeed the real reconstruction effects can be complicated I at least here would like to see some explanation of the potential compromises/challenges in the reconstruction. A lot of the light will even undergo multiple internal reflections and the timing information of the detected photons is afterwards very limited. So also the information from the light, which anyway also without the modification would have been detected, is compromised, as the detector cannot distinguish the origin of the light. This would thus in any case negatively impact the reconstruction.

It is possible that for the goal of measuring high energetic particles the angular resolution of their direction is anyway governed by the detector extension and the timing less crucial, but then for such high energetic events it is not obvious how much information a local detector enlargement very close to an existing detector unit would add. Can the here suggested path actually be compared to the option of higher quantum efficiency detectors with significantly worse time resolution (or e.g. using directly WLS in the DOM)? I would assume that there exist some preliminary studies from IceCube exploring such to compare to? This could also provide guidance how much such loss of timing information could potentially compromise the abilities for background (atmospheric muon) rejection, which is a crucial item.

The study further considers different filling factors of the hole with wavelength shifting additives in the water and comes to the conclusion of a $\sim 94\%$ filling of the hole as rough optimum. Even that it is stressed that this is only an approximate optimum and also many simplifying assumptions contributed here, it is not completely obvious to me what defines the 94% as optimum. Looking at figure 3 I cannot identify a trend other than increasing gains from 0% to 94%, I suggest to also show some curves between 94% and 100% to demonstrate the trend in decreasing the gain enhancement. I expect that this comes from a negative contribution for e.g. 100% filling as then light is rescattered which otherwise would have reached the detector, so the curve for 100% starting at 0 could seem misleading in not showing this negative contribution.

Overall I recommend the paper for publication, but would like to ask that my comments above can be addressed and incorporated.

Kind regards,

Dorothea Samtleben

Associate Professor
Leiden University & Nikhef, Amsterdam

Reviewer #2 (Remarks to the Author):

Dear Authors,

I divided my review into two parts. The first part consists in line-by-line comments and suggestions with the aim to improve the writing of the paper; the second part contains some questions and remarks including my general opinion about the manuscript.

Line-by-line comments

(For convenience, I generated a file with line numbers. I attach this file.)

Authors

If you want to put author names in alphabetical order, you should swap the last ones.

Abstract

- **may increase its sensitivity by an order of magnitude**

Please, check the enhancement of sensitivity that the IceCube-Gen2 Collaboration has estimated for the next-generation detector (see my following comments).

Text

Line 1) “**recently**”

Not really. You used this adverb in 2014 (in reference [2] here), one year after the publication of [1] in Science.

You could add a more recent reference, for example:

Observation and Characterization of a Cosmic Muon Neutrino Flux from the Northern Hemisphere using six years of IceCube data
IceCube Collaboration: M. G. Aartsen *et al.*
Astrophys.J. 833 (2016) no.1, 3
e-Print: [arXiv:1607.08006](https://arxiv.org/abs/1607.08006)

Line 4) “**tenfold**”

It is not what it is written in reference [2], where:

- at page 13, after very interesting and useful considerations on the fact that different factors determine the detector sensitivity to astrophysical neutrinos, we find: “For a traditional **point source** search ... we aim for an increase in sensitivity of a factor of 5 and beyond”;
- at page 9, “A next-generation neutrino observatory with 5 times the **point-source** sensitivity of IceCube and otherwise similar detector performance would increase the sensitivity to **source densities and rates** by about two orders of magnitude”, citing your reference [3];
- at page 10, “Even the simplest extension of IceCube will result in **rates** increased by a factor 5 or more”;

- at page 14, “An extended geometry will yield a factor of 10 increase in **double bang tau neutrino event rates** at PeV energies compared to IceCube”.
- At RICH 2016 (talk by S. Toscano) it was said that the goal for IceCube-Gen2 is: “a factor of 5 more sensitive”.

(https://indico.cern.ch/event/393078/contributions/2195247/attachments/1333519/2005014/1-Toscano_RICH2016.pdf).

- At ICRC 2017 (talk by J. van Santen) it was specified that the point source sensitivity should be 3.5 times better than IceCube at $\delta = 0$, and up to 8 times better at $\delta = -45^\circ$ with a surface veto.

So, please, specify better what you would like to say with “sensitivity” and cite proper references as support.

Line 5) Please, **check references [3] and [4]**.

- In Reference [4] the statement is “IceCube-Gen2 ... is expected to improve neutrino detection rate by a factor of 10”.
- In Reference [3] is claimed an enhancement of 2 orders of magnitude in rates.

Line 12) light can be shifted from UV **wavelengths produced by** Cherenkov radiation -> light can be shifted from UV wavelengths of Cherenkov radiation

Line 13) **optical wavelength** -> visible wavelength

Line 15) **Here we explore the possibility to deploy WLS and light concentration....**

It seems that you want to deploy light concentration.
Please, rephrase this sentence.

Line 18) **see Fig.** -> see Fig. 1

Figure number is missing.

Line 20) **“Results”**

Already?

Line 28 and in following lines) $d_{dom} \rightarrow d_{DOM}$, $h_{dom} \rightarrow h_{DOM}$, and $h_{wls} \rightarrow h_{WLS}$, $A_{wls} \rightarrow A_{WLS}$, and $F_{wls} \rightarrow F_{WLS}$

As you do in Figure 1 for d_{DOM} , h_{DOM} , and h_{WLS} .

Line 32) **the central cylinder of the hole** -> the cylindrical hole

Line 38) In Figure 1 you draw a **right-handed triad**. I suggest continuing to use it. If you want to speak about angles between vectors, you have to take into account their orientation and measure angles counterclockwise.

So, referring to Figure 2 (a), with the y -axis in the opposite direction:

$$\widehat{N}_s = (-\cos \theta_o, -\sin \theta_o, 0)$$

Please, keep attention to the subscript $\theta_o \rightarrow \theta_o$ and it is correct to put the subscripts (“i”, “o” and “s”) for $N, \theta, \alpha, x, y, z$ in roman and not in italic, since they are not symbols of physical variables.

Line 39) where $-\theta_o$ is the angle between \widehat{N}_s and $(\widehat{N}_o)_{xy}$ \rightarrow where θ_o is the angle between $-(\widehat{N}_o)_{xy}$ and \widehat{N}_s

Line 41) **diffracted light beam** \rightarrow refracted light beam

There are no diffraction phenomena here!!

Line 41) **let the angle between \widehat{N}_s and \widehat{N}_o be denoted with α_o , and that between \widehat{N}_s and \widehat{N}_i be denoted with α_i .** \rightarrow let the angle between $-\widehat{N}_o$ and \widehat{N}_s be denoted with α_o , and the angle between \widehat{N}_i and $-\widehat{N}_s$ be denoted with α_i .

Line 46) **diffraction** \rightarrow refraction

Line 47) **We solve this equation system analytically... to obtain... from which we get...**

Equation (3) can be derived simply by

$$(\widehat{N}_i)_{xy} \cdot \widehat{N}_s = -x_i \cos \theta_o - y_i \sin \theta_o = \sqrt{x_i^2 + y_i^2} \cos(\pi + \theta_i) = -\sqrt{x_i^2 + y_i^2} \cos(\theta_i)$$

from which we get

$$\theta_i = \arccos\left(\frac{x_i \cos \theta_o + y_i \sin \theta_o}{\sqrt{x_i^2 + y_i^2}}\right).$$

Equation 3) Please, pay attention to the **square root at the denominator**.

Line 49) **We now calculate the width $W(\varphi_o, n_i)$**

It would be nice to show how W depends on φ_o and n_i .

Line 50) **the drill hole's central cylinder** \rightarrow the drill hole cylinder

Line 51) **by requiring its edges to satisfy $\tan \theta_{i,W} = d_{DOM}/d_{hole}$** \rightarrow by requiring its edges to satisfy $\sin \theta_{i,W} = d_{DOM}/d_{hole}$

Line 51) see Fig. 1 -> see Fig. 2

Line 55) shown in Fig. 1 -> shown in Fig. 3

Line 64) at which IceCube's DOMs have $\epsilon_{\text{eff}} = 40\%$ higher light detection efficiency

This sentence is not clear. In the reference you cite we find that it is possible to increase the light detection efficiency by more than 40 % by using some wavelength shifters: either PPO or Butyl-PBD (see their Table 2). Do you think of using these materials?

Lines 63-68) It would be better if you could rephrase this paragraph, since the first part of point (i) is the same of the first part of point (iii).

Line 69) total internal reflection -> total internal reflection (tir)

In this way, you can use the acronym tir afterwards.

Line 73) the scattering length at 400 nm is comparable to h_{dom} [11]

In [11] we find "The absorption length of light from UV to blue varies between 50 m and 150 m, depending on depth".

Reference [11] is not appropriate, there are no information on scattering length.

Line 74) while the absorption length is much greater, 50 m – 190 m [12]

It is not what is written in [12], where we find "around 2400 m depth, the average effective scattering length is close to 50 m and the average absorption length is close to 190 m. These values are for 400 nm light".

Reference [12] contains information on both scattering and absorption lengths. A scattering length of 50 m (at 2.4 km depth and for 400 nm light) is comparable to $h_{\text{DOM}} = 17$ m? Please, comment about it.

Please, rephrase lines 73-74, and remember that scattering and absorption lengths depend on depth and light wavelength. You can take a look at Ackermann M. *et al.* Optical properties of deep glacial ice at the South Pole, JOURNAL OF GEOPHYSICAL RESEARCH, VOL. 111, D13203, doi:10.1029/2005JD006687, 2006

Line 76) DOM-a fraction

Please, use commas or hyphens.

Line 80) see Fig. -> see Fig. 1

Figure number is missing.

Line 82) this corresponds to an approximate collective area

What “this” is referred to? To the situation in which the drill hole is infused with WLS for a height of h_{WLS} ? Could you rephrase this sentence? What does “collective” mean in this context?

Line 83) 0.25 -> 1/4

It would be more readable.

Equation 5) $F(n_o)$ -> $F_n(n_o)$

Line 90) therefore this contribution will be Eq. 4

Please, rephrase this sentence.

Line 92) enhancement of the DOMs -> enhancement of the DOM sensitivity

Line 94) see Fig. 1 -> see Fig. 3

Line 95) enhancement -> sensitivity enhancement

Line 96) = -> \cong

Line 103) (e.g.,^{13,14} -> (e.g.,^{13,14})

Line 104) 589nm -> 589 nm

Line 131) in water $n=1.38$ at 589nm^{14} , while amyl acetate ($n=1.4$ at 589nm and 20C^{14} freezes at -71C. -> in water $n = 1.38$ at 589 nm^{14} , while amyl acetate ($n = 1.4$ at 589 nm and $20\text{ }^\circ\text{C}^{14}$) freezes at $-71\text{ }^\circ\text{C}$.

Line 144) to be be kept -> to be kept

Line 155) on these factors on IceCube’s photon sensitivity -> of these factors on IceCube’s photon sensitivity

Line 157) Discussion

Maybe, Conclusions?

Line 218) Klaudio Kopper -> Claudio Kopper

Figures

Figure 2

Figure (a) As commented for the Line 38, in Figure 1 you draw a right-handed triad. I suggest continuing to use it. Thus, the y-axis should point to the opposite direction. The z-axis should not be drawn in perspective as the figure is represented in the x-y plane. We see the projections of vectors \widehat{N}_o and \widehat{N}_i in the x-y plane, thus \widehat{N}_o and \widehat{N}_i should be replaced with $(\widehat{N}_o)_{xy}$ and $(\widehat{N}_i)_{xy}$.

Figure (c) The y-axis should point to the opposite direction. For completeness, it would be useful to show the \widehat{N}_s vector.

Caption: analysis -> calculation

Figure 3

y-axis title: DOM enhancement -> DOM sensitivity enhancement

x-axis title: Refractive index -> Inner ice refractive index (n_i)

Caption title: as a function ice refractive index -> as a function of the inner ice refractive index

References

4. Bartos, I. *et al.* Prospects of Establishing the Origin of Cosmic Neutrinos using Source Catalogs. *Phys.Rev. D96 no.2, 023003 (2017)*.

8. Schulte, L. & others -> Schulte, L. *et al.*

11. Ahrens, J. *et al.* -> IceCube Collaboration

Like ref. 1?

12. Abbasi, R. *et al.* -> IceCube Collaboration

14. W. M. H. -> Haynes, W. M.

16. Amram, P. *et al.* -> ANTARES Collaboration

19. Link written twice.

Questions and remarks

- Thanks to an adequate wavelength shifter infusion, and the ice refractive index increase, you expect to enhance the IceCube's photon sensitivity (the sensitivity of DOMs), as you stress at some points in the paper. But you make also several references to the IceCube's sensitivity: the sensitivity of the detector to astrophysical neutrinos. My question is: does an enhancement of the photon sensitivity always cause an enhancement of the detector sensitivity? For each detection channel? For the search of point sources as well as of diffuse sources?
- The speed of light in a material depends on its wavelength. Have you estimated the temporal dispersion of photons resulting from the change of wavelength? Could this temporal dispersion influence the angular reconstruction of neutrino events?
- IceCube-Gen2 Collaboration is studying a prototype of the so-called wavelength-shifting optical module (WOM). Do you think that the solution you propose is better than the WOM? Why?

- In the “Next steps” section, you make a long to-do-list:
 - Line 111: Further studies ... are necessary
 - Line 115: further studies are necessary
 - Line 129: ... must be investigated
 - Line 140: ... will also need to be investigated
 - Line 147: We further need to study...and the feasibility of carrying the needed materials to the South Pole. (Key point)
 - Lines 153-156 Finally, ... will need to be explored (Key point)

Then, at line 164 you write “there is a potentially enormous benefit in modifying the properties of the ice in drill holes during the placement of DOM strings”. This is not obvious if you have the doubt expressed at lines 153-156. In addition, if it would not be possible carrying the needed materials to the South Pole, this will remain a fantasy.

In summary, I think that before the publication of this paper you should find an answer or start to search for an answer to, at least, the two **Key points**. Moreover, you should make the assumptions more realistic. Averaging over the whole sky (line 24) as well as the integration over all vertical angles (line 53) are too simplistic. It is not possible to ignore that the contribution of background from northern and southern sky is different and that also the type of DOM influences the detection performances.

Reviewer #3 (Remarks to the Author):

The paper deals with a likely cost-effective way to significantly increase the efficiency of the ice-based neutrino detector IceCube. I congratulate the authors to this idea, which I find very intriguing. I do believe that the idea and the presented work deserves publishing, but I do not think that Nature Communications is the right journal at this stage.

The idea of using wavelength shifting optics is not new as the authors themselves point out, but the idea to infuse the ice of the drill holes is novel.

IceCube is the only ice-based Cherenkov neutrino detector. While it can be envisaged that volumes with wavelength shifting material could be also constructed in water based Cherenkov telescopes, the use of this techniques seems much less obvious. In consequence, the interest in this paper for the community of neutrino astrophysics is in my view rather limited, whereas it is a very interesting and creative idea for the IceCube collaboration.

The claims that are made in the paper are convincing under the assumptions that are clearly stated. It is probably slightly misleading to claim an increase in the IceCube-Gen2 sensitivity, as the sensitivity does not only depend on the flux of incoming photons. I assume that what the author call "DOM enhancement" is the quantity defined in equation (4). The crucial question is whether the signal to noise is going to be enhanced, which also depends on the reconstruction capability. The authors do allude to this fact themselves. To answer the question will take much more detailed studies on the implementation and effect of performance for IceCube. Attempting to e.g. estimate the increase in point source detection sensitivity under realistic assumption how the increased DOM efficiency will affect reconstruction would certainly make the paper more useful.

=====

Reviewer #1 (Remarks to the Author):

COMMENT: The main gain in sensitivity is demonstrated to stem from the addition of wavelength-shifted light: The rescattering of the light passing the hole is in this way becoming detectable, but otherwise is assumed to be missed in the detection.

Here the paper does not provide guidance to the amount of information loss by rescattered light, which otherwise could actually have been detected elsewhere (I assume though very little with the sparse IceCube Gen2 design). It also does not elaborate how much the scattering compromises the information from the light which then is detected. It is only pointed out that effects on the reconstruction are not yet considered and deserve further detailed studies.

RESPONSE: To understand the role of information loss due to rescattered light not reaching a distant DOM it was headed to, we estimated for a given DOM the fraction of its 4π view covered by a neighboring drill hole at 240m distance. This fraction, assuming the DOM is in the middle of the 1km hole, is $\sim 4.5 \times 10^{-4}$. Even with multiple neighboring holes, this fraction is negligible. Farther holes can essentially be neglected as absorption will mitigate the amount of photons reaching the DOM from them. Even for 240m, we estimate that the fraction of photons reaching a DOM are 10^{-6} and 10^{-8} for track and cascade events, respectively. In any case, we find holes at $2 \times 240\text{m}$ cover roughly $1/4^{\text{th}}$ the sky area of holes at 240m distance. We added a paragraph to the end of the section “Wavelength shifters” to point out that this effect is negligible.

To study the effect of scattering on information loss, we calculated the distribution of temporal time delay for photons arriving at a DOM for an example case of 100m source distance, providing prescription of how this can be done for any source location. We present the obtained distribution in comparison with the distribution expected for direct propagation to the DOM, the so-called Pandel function. We show that compared to the Pandel function there is a characteristic time delay of 100ns for a fiducial $n_i=1.4$, which is significantly smaller than the spread of photons at this distance scale, albeit the first photons, that are especially important in some

reconstructions, are characteristically more delayed in our case. The results are presented in detail in a new section under methods. We do recognize, however, that this is just a first step and the effect of scattering needs to be accounted for in detailed simulations of the detector and event reconstruction.

COMMENT: Even that indeed the real reconstruction effects can be complicated I at least here would like to see some explanation of the potential compromises/challenges in the reconstruction. A lot of the light will even undergo multiple internal reflections and the timing information of the detected photons is afterwards very limited. So also the information from the light, which anyway also without the modification would have been detected, is compromised, as the detector cannot distinguish the origin of the light. This would thus in any case negatively impact the reconstruction.

RESPONSE: As mentioned above, we now include a detailed analysis of the expected additional time delay from infused ice. This time delay indeed widens the distribution for detected photons, which mitigates the precision of the reconstruction of source properties. In addition, the time delay hides the source's direction compared to the DOM and depends on the direction from the infused ice column, further complicating the reconstruction. On the other hand, we will still have direct photons from the source that arrive at the DOM without entering the WLS region for most source directions. In this sense, if one looks at the earliest photons, those will very likely be the direct photons, providing the same information on the source distance as if there was no WLS. In addition, we point out that the time delay due to rescattering will become less important for Cherenkov emission originating at larger distances from a given DOM. Farther emission, which has more limited photon flux and already a large delay due to scattering even without WLS, is the more problematic to reconstruct, and the added WLS photons can be highly beneficial for this category. On the other extreme, for nearby Cherenkov emission the first direct photons can be sufficiently numerous such that we don't need additional WLS photons to reconstruct properties. Therefore, WLS should be helping where it matters. We note, nevertheless, that these general concepts need to be explored numerically. We included a new paragraph at the end of the Next Steps section with this argument.

COMMENT: It is possible that for the goal of measuring high energetic particles the angular resolution of their direction is anyway governed by the detector extension and the timing less crucial, but then for such high energetic events it is not obvious how much information a local detector enlargement very close to an existing detector unit would add. Can the here suggested path actually be compared to the option of higher quantum efficiency detectors with significantly worse time resolution (or e.g. using directly WLS in the DOM)? I would assume that there exist some preliminary studies from IceCube exploring such to compare to? This could also provide guidance how much such loss of timing information could potentially compromise the abilities for background (atmospheric muon) rejection, which is a crucial item.

RESPONSE: This is an important point, indeed in effect our proposal is equivalent to significantly increasing DOM sensitivity directly, or to significantly decreasing the DOM separation, with the added timing uncertainty in our case. We emphasize here that an improved DOM and infused ice

parallelly increase the total DOM sensitivity. For instance if we increase DOM effective area by X%, and infused ice improves DOM sensitivity by Y%, then the total improvement should be $(1+X\%/100) * (1+Y\%/100)$. For WOM the situation is somewhat different as the incoming photons from the infused ice are already higher wavelength, so they will propagate differently within the WOM and the added efficiency we count in from the wavelength transfer will not apply. We added a new bullet point on these possibilities and limitations to the Next Steps section.

Regarding the overall benefits of e.g. WOM vs standard DOMs, the photon detection efficiency had been investigated in detail by IceCube. However, to our knowledge, there has been no systematic study of how detector sensitivity to, say, point sources improves upon changing DOMs. One, albeit limited, exception, is the comparison of different geometries for PINGU, where multiple geometries were compared where the inter-DOM spacing was modified, although along with string distances. We present the summary of this comparison in our Methods section (subsection Sensitivity estimates), where we use it to argue that the increase in DOM sensitivity, along with other changes such as in string spacing, can introduce significant gains to search sensitivity (in this example to sensitivity to mass ordering specifically, but likely to other measures as well).

We note here that some studies are under way to characterize what the Referee suggests, but at this point they are very preliminary, even though they will be extremely useful not just for the present work but for understanding the best upgrade possibilities as well.

COMMENT: The study further considers different filling factors of the hole with wavelength shifting additives in the water and comes to the conclusion of a ~94% filling of the hole as rough optimum. Even that it is stressed that this is only an approximate optimum and also many simplifying assumptions contributed here, it is not completely obvious to me what defines the 94% as optimum. Looking at figure 3 I cannot identify a trend other than increasing gains from 0% to 94%, I suggest to also show some curves between 94% and 100% to demonstrate the trend in decreasing the gain enhancement. I expect that this comes from a negative contribution for e.g. 100% filling as then light is rescattered which otherwise would have reached the detector, so the curve for 100% starting at 0 could seem misleading in not showing this negative contribution.

RESPONSE: Indeed the trend towards the optimum was not obvious from Fig. 3, where we tried to limit the number of curves for clarity. We now added an inset to Fig. 3 that shows DOM sensitivity enhancement as a function of % filling for a fiducial $n_i=1.4$. This shows that a curve has a clear optimum, and at the same time the enhancement is not overly sensitive to the filling % -- a few percent change will keep DOM enhancement within a few % of the optimum. As the Referee points out, the optimum is determined by two competing effects. On the one hand, higher % filling means more rescattered light by the WLS that can be detected by the DOM. On the other hand, WLS blocks the direct path of light to the DOM, which is especially important for the region around the DOM. We added a sentence to section "DOM sensitivity enhancement" to explain this.

=====

Reviewer #2 (Remarks to the Author):

Authors

COMMENT: Authors If you want to put author names in alphabetical order, you should swap the last ones.

RESPONSE: The order follows the amount of contribution to the manuscript.

Abstract

COMMENT: may increase its sensitivity by an order of magnitude Please, check the enhancement of sensitivity that the IceCube-Gen2 Collaboration has estimated for the next-generation detector (see my following comments).

RESPONSE: The Referee raises the very valid point of needing a clearer definition of sensitivity, in this and some of the following comments, including one in the questions and remarks. This is especially important because IceCube and similar detectors aim to detect multiple kinds of signals, such as neutrinos at different energies, or individual sources versus populations, and sensitivity can be defined separately for each category. We aim to address this concern in general by clarifying in the text our definitions when necessary, and making the references to sensitivity qualitative, instead of quantitative, whenever we are emphasizing next generation instruments. Please find further details in our responses to individual comment on sensitivity in the following.

Sensitivity increase for Gen2 will be largely dependent on the type of search we consider, and in addition the factor of 10 increase we quoted previously is not guaranteed at this point. We originally wrote this because this was the target for the upgrade for point source searches, but it now seems that something more modest may be more realistic, although current estimates and design considerations are still in flux. We modified the sentence to give a qualitative, and not quantitative, statement of the improvement.

Text

COMMENT:

Line 1) “recently” Not really. You used this adverb in 2014 (in reference [2] here), one year after the publication of [1] in Science. You could add a more recent reference, for example: Observation and Characterization of a Cosmic Muon Neutrino Flux from the Northern Hemisphere using six years of IceCube data IceCube Collaboration: M. G. Aartsen *et al.* *Astrophys.J.* 833 (2016) no.1, 3 e-Print: arXiv:1607.08006

RESPONSE: We modified the sentence to specify that the diffuse neutrino flux has been observed since 2013. We now additionally cite the manuscript the Referee suggested.

COMMENT: Line 4) “tenfold”

It is not what it is written in reference [2], where:

- at page 13, after very interesting and useful considerations on the fact that different factors determine the detector sensitivity to astrophysical neutrinos, we find: “For a traditional **point source** search ... we aim for an increase in sensitivity of a factor of 5 and beyond”;
- at page 9, “A next-generation neutrino observatory with 5 times the **point-source** sensitivity of IceCube and otherwise similar detector performance would increase the sensitivity to **source densities and rates** by about two orders of magnitude”, citing your reference [3];
- at page 10, “Even the simplest extension of IceCube will result in **rates** increased by a factor 5 or more”;
- at page 14, “An extended geometry will yield a factor of 10 increase in **double bang tau neutrino** event **rates** at PeV energies compared to IceCube”.
- At RICH 2016 (talk by S. Toscano) it was said that the goal for IceCube-Gen2 is: “a factor of 5 more sensitive”.

(https://indico.cern.ch/event/393078/contributions/2195247/attachments/1333519/2005014/1-Toscano_RICH2016.pdf).

- At ICRC 2017 (talk by J. van Santen) it was specified that the point source sensitivity should be 3.5 times better than IceCube at $\delta = 0$, and up to 8 times better at $\delta = -45^\circ$ with a surface veto. So, please, specify better what you would like to say with “sensitivity” and cite proper references as support.

RESPONSE: We changed the text to specify that we are referring to the detector’s sensitivity to point sources, and to state that the improvement will be a “factor of 5 and beyond”, in line with our reference 3 (note that it was previously reference 2). While this “factor of 5” depends on the details of the point source analysis considered for establishing the improvement, we find quoting this improvement from IceCube’s overview paper on the upgrade the most informative to the reader in order to give a sense of the goals to the reader. We reference the overview paper for further details.

COMMENT: Line 5) Please, check references [3] and [4].

- In Reference [4] the statement is “IceCube-Gen2 ... is expected to improve neutrino detection rate by a factor of 10”.
- In Reference [3] is claimed an enhancement of 2 orders of magnitude in rates.

RESPONSE: This is another example for when sensitivity is defined differently for two studies. Both works consider the sensitivity of IceCube-Gen2 to probe the populations of sources with different number densities. However, Ref [3] considers identification at 90% confidence level, while Reference [4] requires 3sigma. The detector’s ability to identify a source population scales

differently for these two distinct limits, hence the different results. Both results, however, are consistent with our statement that for typically considered source populations, the proposed upgrade is expected to be (likely) sufficient to identify the population.

COMMENT: Line 12) light can be shifted from UV wavelengths produced by Cherenkov radiation
-> light can be shifted from UV wavelengths of Cherenkov radiation

RESPONSE: corrected.

COMMENT: optical wavelength -> visible wavelength

RESPONSE: corrected.

COMMENT: Line 15) Here we explore the possibility to deploy WLS and light concentration....
It seems that you want to deploy light concentration. Please, rephrase this sentence.

RESPONSE: we clarified the sentence.

COMMENT: Line 18) see Fig. -> see Fig. 1 Figure number is missing.

RESPONSE: corrected.

COMMENT: Line 20) "Results" Already?

RESPONSE: As we understand this naming order is the requirement for Nature Commun.

COMMENT: Line 28 and in following lines) d_{dom} -> $dDOM$, h_{dom} -> $hDOM$, and $hwls$ -> $hWLS$,
 $Awls$ -> $AWLS$, and $Fwls$ -> $FWLS$ As you do in Figure 1 for $dDOM$, $hDOM$, and $hWLS$.

RESPONSE: implemented.

COMMENT: Line 32) the central cylinder of the hole -> the cylindrical hole

RESPONSE: there are two cylinders we refer to, one which is the hole itself and one imaginary cylinder with the diameter of the DOMs. Here we are referring to the latter. We clarified the text to make it less ambiguous.

COMMENT: Line 38) In Figure 1 you draw a right-handed triad. I suggest continuing to use it. If you want to speak about angles between vectors, you have to take into account their orientation and measure angles counterclockwise.

So, referring to Figure 2 (a), with the y-axis in the opposite direction:

$N_s = -\cos\theta_o, -\sin\theta_o, 0$ Please, keep attention to the subscript θ , -> θ_o and it is correct to put the subscripts ("i", "o" and

“s”) for N , θ , α , x , y , z in roman and not in italic, since they are not symbols of physical variables.

RESPONSE: We adopted a right-handed triad everywhere. We modified N_s accordingly as indicated by the Referee, and fixed the subscripts.

COMMENT: Line 39) where $-\theta_0$ is the angle between N_s and N_o ./ -> where θ_0 is the angle between $-N_o$./and N_s

RESPONSE: modified.

COMMENT: Line 41) diffracted light beam -> refracted light beam There are no diffraction phenomena here!!

RESPONSE: corrected.

COMMENT: Line 41) let the angle between N_s and N_o be denoted with α_o , and that between N_s and N_i be denoted with α_i .-> let the angle between $-N_o$ and N_s be denoted with α_o , and the angle between N_i and $-N_s$ be denoted with α_i .

RESPONSE: modified.

COMMENT: Line 46) diffraction -> refraction

RESPONSE: corrected.

COMMENT: Line 47) We solve this equation system analytically... to obtain... from which we get... Equation (3) can be derived simply by

$$N_i \cdot N_s = -x_3 \cos \theta_0 - y_3 \sin \theta_0 = x_{i5} + y_{i5} \cos \pi + \theta_i = -x_{i5} + y_{i5} \cos \theta_i$$

from which we get

$$\theta_i = \arccos \frac{x_3 \cos \theta_0 + y_3 \sin \theta_0}{x_{i5} + y_{i5}}$$

Equation 3) Please, pay attention to the square root at the denominator.

RESPONSE: While Eq 4 (formerly Eq 3) can indeed be obtained simply as the Referee suggests, the equation system we present is needed to obtain x_i , y_i and z_i . We omit the detailed derivation in the manuscript as it is technical but straightforward.

To clarify the description, we added a new equation, Eq. 3, which adds the Referee’s simple derivation for θ_i , noting that x, y, z in this equation were obtained already using the equation system.

We also corrected the square root the Referee caught. We note that this error was present in the manuscript but not in our numerical results.

COMMENT: Line 49) We now calculate the width $W(\varphi_o, n_i)$ It would be nice to show how W depends on φ_o and n_i .

RESPONSE: We can only calculate W in the general case numerically, but this way we find that it is linearly dependent of n_i to good approximation independently of φ_o until it reaches $W=d_{\text{hole}}$. For the special case of $\varphi_o=0$, it is straightforward to calculate the relation analytically, it is $W = d_{\text{hole}} * n_i / n_o$. We added text to describe these two cases. We copy here a figure showing the dependence of W on n_i and φ_o (in legend). We did not include this figure in the manuscript.

COMMENT: Line 50) the drill hole's central cylinder -> the drill hole cylinder

RESPONSE: We need to differentiate here between the full drill hole and the central part that has the width of a DOM. To clarify this in the text, we modified the text highlighted by the Referee to “central cylinder in the drill hole with diameter d_{DOM} ”.

COMMENT: Line 51) by requiring its edges to satisfy $\tan\theta_i, W = d_{\text{DOM}}/d_{\text{hole}}$ -> by requiring its edges to satisfy $\sin\theta_i, W = d_{\text{DOM}}/d_{\text{hole}}$

RESPONSE: We corrected this typo. Our numerical calculations used the correct formula.

COMMENT: Line 51) see Fig. 1 -> see Fig. 2

RESPONSE: Corrected.

Line 55) shown in Fig. 1 -> shown in Fig. 3

RESPONSE: Corrected.

COMMENT: Line 64) at which IceCube's DOMs have $\epsilon_{\text{eff}} = 40\%$ higher light detection efficiency
This sentence is not clear. In the reference you cite we find that it is possible to increase the light detection efficiency by more than 40% by using some wavelength shifters: either PPO or Butyl-PBD (see their Table 2). Do you think of using these materials?

RESPONSE: We take this 40% increase as a fiducial value for the effect of changing the wavelength from UV to optical by the WLS. We do not aim to specify at this point the material used as WLS, but want to account for the change in detection efficiency, and adopt these values as fiducial. We clarified the text to separate the assumption that WLS changes wavelength from UV to optical, and that we additionally assume that this change will result in a 40% fiducial change in detection efficiency.

COMMENT: Lines 63-68) It would be better if you could rephrase this paragraph, since the first part of point (i) is the same of the first part of point (iii).

RESPONSE: We reordered and rephrased the paragraph to avoid repetition and also better address the previous comment.

COMMENT: Line 69) total internal reflection -> total internal reflection (tir)
In this way, you can use the acronym tir afterwards.

RESPONSE: We only use this term twice so it is probably easier to keep it spelled out.

COMMENT: Line 73) the scattering length at 400 nm is comparable to h_{DOM} [11]
In [11] we find "The absorption length of light from UV to blue varies between 50 m and 150 m, depending on depth".
Reference [11] is not appropriate, there are no information on scattering length.

RESPONSE: We updated the reference to "Measurement of South Pole ice transparency with the IceCube LED calibration system" by IceCube which details the measurement of scattering length as a function of depth.

COMMENT: Line 74) while the absorption length is much greater, 50 m – 190 m [12]
It is not what is written in [12], where we find "around 2400 m depth, the average effective scattering length is close to 50 m and the average absorption length is close to 190 m. These values are for 400 nm light".
Reference [12] contains information on both scattering and absorption lengths. A scattering length of 50 m (at 2.4 km depth and for 400 nm light) is comparable to $h_{\text{DOM}} = 17$ m? Please, comment about it.
Please, rephrase lines 73-74, and remember that scattering and absorption lengths depend on depth and light wavelength. You can take a look at Ackermann M. *et al.* Optical properties of deep glacial ice at the South Pole, JOURNAL OF GEOPHYSICAL RESEARCH, VOL. 111, D13203, doi:10.1029/2005JD006687, 2006

RESPONSE: We corrected the absorption length quote, and rephrased the paragraph to make it clearer what the intention is and what we actually do. We want to account for scattering and absorption with a single model in which we don't let the photons travel past a DOM, making their maximum path length $\sim h_{\text{DOM}}=17\text{m}$. We argue that this is comparable or shorter than the scattering length, and shorter than the absorption length, so it can be considered a conservative assumption. We further added the citation suggested by the Referee.

COMMENT: Line 76) DOM—a fraction Please, use commas or hyphens.

RESPONSE: corrected.

COMMENT: Line 80) see Fig. -> see Fig. 1 Figure number is missing.

RESPONSE: corrected.

COMMENT: Line 82) this corresponds to an approximate collective area
What "this" is referred to? To the situation in which the drill hole is infused with WLS for a height of h_{WLS} ? Could you rephrase this sentence? What does "collective" mean in this context?

RESPONSE: We rewrote the corresponding text and added a definition to explain better what we mean. "Collective" was a typo, we meant "collecting," but probably a better term is "effective," given that we assume 100% WLS efficiency. We adopted this term.

COMMENT: Line 83) 0.25 -> 1/4
It would be more readable.

RESPONSE: modified.

COMMENT: Equation 5) $F(no)$ -> $F_n(no)$

RESPONSE: corrected.

COMMENT: Line 90) therefore this contribution will be Eq. 4 Please, rephrase this sentence.

RESPONSE: We clarified the sentence.

COMMENT: Line 92) enhancement of the DOMs -> enhancement of the DOM sensitivity

RESPONSE: Modified.

COMMENT: Line 94) see Fig. 1 -> see Fig. 3

RESPONSE: corrected.

COMMENT: Line 95) enhancement -> sensitivity enhancement

RESPONSE: corrected.

COMMENT: Line 96) = -> \cong

RESPONSE: modified.

COMMENT: Line 103) (e.g., 13, 14 -> (e.g., 13, 14)

RESPONSE: corrected.

COMMENT: Line 104) 589nm -> 589 nm

RESPONSE: Corrected.

COMMENT: Line 131) in water $n=1.38$ at 589nm ¹⁴, while amyl acetate ($n=1.4$ at 589nm and 20°C ¹⁴ freezes at -71°C . -> in water $n = 1.38$ at 589 nm ¹⁴, while amyl acetate ($n = 1.4$ at 589 nm and 20°C ¹⁴) freezes at -71°C .

RESPONSE: Corrected.

COMMENT: Line 144) to be be kept -> to be kept

RESPONSE: Corrected.

COMMENT: Line 155) on these factors on IceCube's photon sensitivity -> of these factors on IceCube's photon sensitivity

RESPONSE: Corrected.

COMMENT: Line 157) Discussion Maybe, Conclusions?

RESPONSE: Our understanding is that this is the standard title for Nature.

COMMENT: Line 218) Klaudio Kopper -> Claudio Kopper

RESPONSE: Corrected.

Figures

COMMENT: Figure 2 Figure (a) As commented for the Line 38, in Figure 1 you draw a right-handed triad. I suggest continuing to use it. Thus, the y-axis should point to the opposite direction. The z-axis should not be drawn in perspective as the figure is represented in the x-y plane. We see the projections of vectors N_o and N_i in the x-y plane, thus N_o and N_i should be replaced with $(N_o)_/$ and $(N_o_{-})_/$.

RESPONSE: We modified both coordinates to right-handed triad as suggested by the Referee. We included the (xy) projection notation to where it is applicable. We did keep the z axis as it indicates the upward direction. While this can be deduced from comparing the xy orientation to Fig 1, we think that this is probably more straightforward to some readers to put things in perspective.

COMMENT: Figure (c) The y-axis should point to the opposite direction. For completeness, it would be useful to show the N_s vector.

RESPONSE: We turned the y axis and added N_s .

COMMENT: Caption: analysis -> calculation

RESPONSE: Modified.

COMMENT: Figure 3

y-axis title: DOM enhancement -> DOM sensitivity enhancement

x-axis title: Refractive index -> Inner ice refractive index (n_i)

RESPONSE: Both implemented.

COMMENT: Caption title: as a function ice refractive index -> as a function of the inner ice refractive index

RESPONSE: Modified.

References

COMMENT: 4. Bartos, I. *et al.* Prospects of Establishing the Origin of Cosmic Neutrinos using Source Catalogs. Phys.Rev. D96 no.2, 023003 (2017).

RESPONSE: corrected.

COMMENT: 8. Schulte, L. & others -> Schulte, L. *et al.*

RESPONSE: Corrected.

COMMENT: 11. Ahrens, J. *et al.* -> IceCube Collaboration Like ref. 1?

RESPONSE: Corrected.

COMMENT: 12. Abbasi, R. *et al.* -> IceCube Collaboration

RESPONSE: Corrected.

COMMENT: 14. W. M. H. -> Haynes, W. M.

RESPONSE: Corrected.

COMMENT: 16. Amram, P. *et al.* -> ANTARES Collaboration

RESPONSE: Corrected.

COMMENT: 19. Link written twice.

RESPONSE: Corrected.

Questions and remarks

COMMENT: Thanks to an adequate wavelength shifter infusion, and the ice refractive index increase, you expect to enhance the IceCube's photon sensitivity (the sensitivity of DOMs), as you stress at some points in the paper. But you make also several references to the IceCube's sensitivity: the sensitivity of the detector to astrophysical neutrinos. My question is: does an enhancement of the photon sensitivity always cause an enhancement of the detector sensitivity? For each detection channel? For the search of point sources as well as of diffuse sources?

RESPONSE:

COMMENT: The speed of light in a material depends on its wavelength. Have you estimated the temporal dispersion of photons resulting from the change of wavelength? Could this temporal dispersion influence the angular reconstruction of neutrino events?

RESPONSE: The refractive index of ice depends relatively little on the wavelengths considered here, and will be a minor contributor to time dispersion. A more important contributor will be scattering along the path towards the DOM, and the different paths between direct photons and photons that first enter the infused ice and then propagate to the DOM within the drill hole. In order to better understand this effect, we carried out a detailed calculation of the residual time distribution for the two cases. We added a new subsection in Results to summarize the

characteristic delay we expect in the infused ice compared to direct photons, and present detailed results, as well as an example distribution, in a new subsection under Methods. The upshot is that we expect order 100ns delay for WLS photons compared to direct photons. This is comparable to typical delays for a source tens of meters or more away from the DOM, and does not affect the distribution of delays substantially. However, we note that timing is a very important element of reconstructing a signal and its parameters, so this delay needs to be incorporated in simulations of neutrino reconstruction. In the present work we did not go beyond presenting the resulting distributions and commenting on their role, but this will be important to investigate in detail in the future.

COMMENT: IceCube-Gen2 Collaboration is studying a prototype of the so-called wavelength-shifting optical module (WOM). Do you think that the solution you propose is better than the WOM? Why?

RESPONSE: The proposed WOM module's usefulness is largely independent of our proposed infused ice model. In fact, if one used infused ice in combination with a WOM, the WOM would increase the efficiency of direct photons, which we treat as a separate source of input, but would not affect the added efficiency from the infused ice. We added a paragraph to Next Steps to highlight this independence and added value to the reader.

COMMENT: In the "Next steps" section, you make a long to-do-list:

- o Line 111: Further studies ... are necessary
- o Line 115: further studies are necessary
- o Line 129: ... must be investigated
- o Line 140: ... will also need to be investigated
- o Line 147: We further need to study...and the feasibility of carrying the needed materials to the South Pole. (Key point)
- o Lines 153-156 Finally, ... will need to be explored (Key point)

Then, at line 164 you write "there is a potentially enormous benefit in modifying the properties of the ice in drill holes during the placement of DOM strings". This is not obvious if you have the doubt expressed at lines 153-156. In addition, if it would not be possible carrying the needed materials to the South Pole, this will remain a fantasy.

In summary, I think that before the publication of this paper you should find an answer or start to search for an answer to, at least, the two Key points.

RESPONSE: We consider the present report as a first step in an interesting new direction of modifying the properties of ice in order to enhance detector sensitivity. Hence, as the Referee rightly points out, there are a number of important open questions to investigate. To further outline the two key points as the Referee suggested, we made the following additions.

Key point 1 – feasibility of carrying the needed material to the South Pole. In order to expand on this point, we created a new subsection under Methods that focuses on additives. We moved the relevant description here from the Next Steps subsection, which we expanded with an analysis

on the amount of additives needed, also providing a feasibility estimate by comparing the amount needed to the amount of cargo+fuel annually delivered to the Amundsen–Scott South Pole Station. We further identify directions in which the required payload can be mitigated further. We believe that these numbers show that we are not in the fantasy category, while we fully agree that further research is important before any full-scale implementation.

Key point 2 – sensitivity, simulations. This is another important point that the Referee raised and that we aimed to incorporate throughout the manuscript. There are a large number of possible sensitivity measures, and it is important to emphasize that one measure does not present a complete picture, and for the final product of IceCube it is important to convert DOM sensitivity to other measures. Such a comparison can better motivate the utility of additives in IceCube’s future upgrades and potentially other detectors, while they will be important in determining the specifics of the application of additives.

In order to start this study, we investigated two particular cases in which we estimated the expected increase for other sensitivity measures. We detail these findings in the methods section and briefly refer to them in the Next Steps section. In one of our examples, we look at supernova neutrinos, which represent an effective increase in the noise level of the detector, and can be identified without a complex reconstruction algorithm. We determine the relationship between how far a supernova can be detected through MeV neutrinos, and DOM sensitivity. We note here that this distance itself is not necessarily the best characterization of the improvement, as beyond the Milky Way the rate of supernovae drops until 3Mpc that is too far with current and near future instruments. Nevertheless, this gives a sense of what improvement we can expect, and as we emphasize for a more local supernova the increased sensitivity to supernova neutrinos can further enhance the information we can collect from one local event.

The second case we study is that of PINGU. For this low energy extension, detailed simulations were carried out for multiple detector geometries, where one of the variables was DOM spacing, which is effectively the same as DOM sensitivity, allowing us to estimate the improvement expected based on the comparison between two geometries. The advantage of this comparison is that detailed simulations and optimization studies have been carried out to find two comparably sensitive geometries, where sensitivity was measured for neutrino mass ordering. The downside is that from this comparison we cannot necessarily extrapolate how further changes in DOM sensitivity would change other sensitivities, or how the time delay introduced by the infused ice would affect sensitivity, nevertheless the comparison shows that a change in DOM sensitivity can indeed have significant effect on other sensitivities as well.

COMMENT: Moreover, you should make the assumptions more realistic. Averaging over the whole sky (line 24) as well as the integration over all vertical angles (line 53) are too simplistic. It is not possible to ignore that the contribution of background from northern and southern sky is different and that also the type of DOM influences the detection performances.

RESPONSE: We added an analysis and a figure on the angular dependence of DOM sensitivity enhancement. The description is pointed to in the DOM sensitivity enhancement subsection, and

the details are presented in the Methods section under subsection Direction-dependent DOM sensitivity enhancement. We show here that there is indeed a variation of the enhancement as a function of the vertical angle of the light source, with the infused ice scenario favoring lower angles, while the pure refractive index scenario has higher enhancement closer to vertical photons, all in line with our expectations. We briefly discuss that this direction dependence does not directly convert to the same direction dependence for source sensitivity, as the direction of emitted Cherenkov photons will not be the same as the neutrino direction. In particular, for cascade events, the photons' direction will be determined primarily by where the neutrino interacted in or around the detector, therefore we can expect emission in all directions. We further speculate photons emitted closer to horizontal will be more relevant as they will likely reach the drill holes at higher flux.

Reviewer #3 (Remarks to the Author):

COMMENT: IceCube is the only ice-based Cherenkov neutrino detector. While it can be envisaged that volumes with wavelength shifting material could be also constructed in water based Cherenkov telescopes, the use of this techniques seems much less obvious. In consequence, the interest in this paper for the community of neutrino astrophysics is in my view rather limited, whereas it is a very interesting and creative idea for the IceCube collaboration.

RESPONSE: Our primary target is indeed IceCube and its upgrades. Nevertheless, we do consider the idea interesting and novel for other scenarios as well. For example water-based Cherenkov detectors are a potential target, where instead of ice one can use additives in water with appropriate containers to keep the additives in the required geometry. Alternatively, one can prefabricate WLS material, e.g., in plastic, that can be placed in the water at positions where it enhances the desired sensitivity. As a second interesting direction, detectors other than IceCube can be envisioned in ice, which can be significantly more cost effective to manufacture than if complete devices were needed to be brought to places like the South Pole. We added a brief description of these possibilities to the "Next Steps" section where we point out that these other possibilities will be worth exploring.

COMMENT: The claims that are made in the paper are convincing under the assumptions that are clearly stated. It is probably slightly misleading to claim an increase in the IceCube-Gen2 sensitivity, as the sensitivity does not only depend on the flux of incoming photons. I assume that what the author call "DOM enhacnemet" is the quantity defined in equation (4). The crucial question is whether the signal to noise is going to be enhanced, which also depends on the reconstruction capability. The authors do allude to this fact themselves. To answer the question will take much more detailed studies on the implementation and effect of performance for IceCube. Attempting to e.g. estimate the increase in point source detection sensitivity under realistic assumption how the increased DOM efficiency will affect reconstruction would certainly make the paper more useful.

RESPONSE: The Referee raises an important point. We modified the text throughout the manuscript to make it clearer that there are widely different measures of sensitivity, and what measure of sensitivity we are referring to at various places. In particular, as the Referee points out, we are primarily focusing on sensitivity to the flux of incoming photons. For a fuller understanding of the potential of the proposed method, and for optimizing its application, it is useful to convert this measure to others, such as the sensitivity to a neutrino point source. Remarkably, to our knowledge and after consulting a large number of colleagues within IceCube, there are currently no systematic studies for water/ice Cherenkov detectors on the dependence of sensitivity (e.g. to point sources) on parameters such as DOM sensitivity to photons. In some cases several scenarios are compared, which is already very useful, but more detailed studies will be highly helpful with upgrades.

Nevertheless, existing studies for PINGU are useful for our case as they demonstrate that the increase of the number of DOMs per string, which is largely the same as increasing DOM sensitivity, can significantly improve the effectiveness of the detector. For PINGU, this is specifically demonstrated for the case of determining mass ordering, but the effectiveness likely applies similarly to other measures. We added a short summary of these studies and their relevance to our case in the Methods section. We further added another simple example, that of MeV supernova neutrinos, for which case the sensitivity improvement due to infused ice is straightforward to quantify.

We additionally added a detailed description of the expected time delay distribution for our case, and compared that to the case with no additives. Timing will play an important role in event reconstruction, and simulations of reconstructions with infused ice will need to take this into account additionally to other simulations that simply assume the improvement of DOM sensitivity. We plan to continue the present work with such detailed simulations that are needed not just for this project but for detector upgrades in general.

REVIEWERS' COMMENTS:

Reviewer #1 (Remarks to the Author):

I appreciate that my concerns were addressed with further investigations and elaborated on and have no objections further for a publication.

Clearly there are several items to be further investigated in detail for a potential implementation, still the investigations in this paper already provide a good motivation to look further into this path to improve the detector.

Small comments: I assume that the authors refer to figure 4 (not figure 1) when pointing at the effect of the time residuals, this still should be changed.

It could also be instructive to show the difference at a closer distance ($\sim 20\text{m}$) where I would expect a less broad distribution.

As for the sensitivity improvement for PINGU: I understand that the efficiency increase will be highly valuable, still, especially for the determination of the mass hierarchy one of the major factors is how well muon neutrino and electron/tau neutrino induced events (tracks vs showers) can be distinguished. I can imagine that for this distinction in the low energetic events (relevant for the determination of the mass hierarchy) the additional loss of timing information could be more critical than in the case of the high energetic events. I appreciate the caveat about conclusions on the mass hierarchy sensitivity in the text, but might also explicitly mention the criticality and challenge of track/shower identification to be considered.

Reviewer #2 (Remarks to the Author):

Dear Authors,

Congratulations for the great work that you made to improve the paper. I have some minor suggestions.

Line-by-line comments (For convenience, I generated a file with line numbers. I attach this file.)

Text

Line 30) the hole be n_o -> the hole be n_i

Line 33, 46, 52) d_{dom} -> d_{DOM}

Line 33) **of** the hole's axis -> the hole's axis

Line 38) Fig. 1

Maybe, Fig. 2 too.

Line 40) where θ_o is the angle between \widehat{N}_s and $-(\widehat{N}_o)_{xy}$ -> where θ_o is the angle between $-(\widehat{N}_o)_{xy}$ and \widehat{N}_s

Line 44) where θ_i is the angle between $-\widehat{N}_s$ and $(\widehat{N}_i)_{xy}$ -> where θ_i is the angle between $(\widehat{N}_i)_{xy}$ and $-\widehat{N}_s$

Line 46) N_i -> N_i

Equation 1) α_i -> α_i

Line 55) $W(\varphi_o = 0; n_i) = d_{\text{DOM}} n_i n_o^{-1}$ -> $W(\varphi_o = 0; n_i) = d_{\text{hole}} n_i n_o^{-1}$

As you wrote in the answers to referees.

Line 64, 68, 71, 74, 75, 82, 86, 143) re-emitting, re-emitted, re-emission with hyphen, but without in the rest of the text. Please, choose one spelling.

Line 69) c.f. -> cf.

Line 73) total internal reflection -> **total internal reflection (tir)**

My suggestion is to add "(tir)" after. In this way, formulas at Line 74 and Equation 6 are clearer.

Line 91) horizontal angle -> vertical angle

As you already did in the rest of the text.

Line 91, 129, 187, 191, 215, Figure 4 caption) source -> light source or Cherenkov source (as you did at Line 192).

It would be better to specify because usually the noun “sources” is used to indicate the astrophysical neutrino sources, as you always did until Line 91.

Line 117) Method -> Methods

Line 144) treat direct -> treat **the** direct

Line 148) with WLS material -> with **a** WLS material

Line 169) you could add citation to Reference 9.

Line 176) pole -> **P**ole

Line 212) neutrino-interaction -> neutrino interaction

Line 212) distances -> distances **from the DOM**

Line 213) neutrino interaction that injects

Neutrino interactions do not inject Cherenkov light. The Cherenkov radiation is induced by the “superluminal” charged particles from neutrino interactions, travelling through the transparent ice, and it is produced by the ice molecules near the tracks of the charged particles that, after being polarized, come back to their initial status. Please, use another verb, like “induce”.

Line 215) origo -> origin

Line 220) to their unscattered **duration** -> to the unscattered time

Line 224, 245, 246) Fig. **1** -> Fig. 4

Line 233) $\alpha_i > \theta_c$

I found unpleasant the use of α_i here, because this symbol was already used at Line 43 (and in Fig. 2) with a different meaning.

If here α_i is the angle between the normal vector at the hole surface and the incident re-emitted (randomly oriented) photon, the photons that have to be discarded since not internally reflected are those with $\alpha_i < \theta_c$.

Line 234 and Equation 8) Please, check the term $z_{WLS} n_i c^{-1} \cos \alpha_i$

Line 240) m -> m

Line 243) 0m -> 0 m

Line 283 - 285) For example, for 50% ethylene glycol in water ...

Please, check the parentheses.

Line 287) an example from above

Could you better clarify?

Line 344) compared to a DOM

Could you better clarify?

Line 347) Fig.2 -> Fig.3

Line 358) emit -> induce

Line 361, 362, 364, 367 and y-axis title in Fig.5) DOM enhancement -> DOM sensitivity enhancement

Figures

Figure 2

(a)

RESPONSE: We modified both coordinates to right-handed triad as suggested by the Referee. We included the (xy) projection notation to where it is applicable. We did keep the z axis as it indicates the upward direction. While this can be deduced from comparing the xy orientation to Fig 1, we think that this is probably more straightforward to some readers to put things in perspective.

If you want to put things in perspective, it is better to include the (xy) notation to \widehat{N}_0 and \widehat{N}_i .

(b)

RESPONSE: We ... added \widehat{N}_s .

\widehat{N}_s not added. Probably this is not the final version of the figure.

Figure 4

Caption) for comparison, we show the distribution for direct propagation modified by the characteristic delay ... (dotted line)

References

8. Goobar, J. & Rodríguez, M. -> Goobar, A. & Rodríguez Martino J.

15. Ahrens, J. *et al.* -> AMANDA Collaboration

18. Aartsen, M.G. *et al.* -> IceCube Collaboration

28. Abbasi, R. *et al.* -> IceCube Collaboration

3., 6., and 9. Please, decide whether to write "arXiv" as here or not as in the rest of the references.

Reviewer #3 (Remarks to the Author):

I am content with how the authors addressed my comments.